# Study on Winding Forming Process of Glass Fiber Composite Pressure Vessel

**DOI:** 10.3390/ma18112485

**Published:** 2025-05-26

**Authors:** Run Wu, Wenlei Zeng, Fangfang Li, Haobin Tian, Xuelei Li

**Affiliations:** School of Intelligent Manufacturing and Control Engineering, Shanghai Polytechnic University, Shanghai 201209, China; 20231510062@sspu.edu.cn (R.W.); hbtian@sspu.edu.cn (H.T.); xlli@sspu.edu.cn (X.L.)

**Keywords:** composite pressure vessel, glass fiber reinforcement, filament winding, blow molding, finite element analysis, structural failure, experimental validation

## Abstract

Composite pressure vessels offer significant advantages over traditional metal-lined designs due to their high strength-to-weight ratio, corrosion resistance, and design flexibility. This study investigates the structural design, winding process, finite element analysis, and experimental validation of a glass fiber-reinforced composite low-pressure vessel. A high-density polyethylene (HDPE) liner was designed with a nominal thickness of 1.5 mm and manufactured via blow molding. The optimal blow-up ratio was determined as 2:1, yielding a wall thickness distribution between 1.39 mm and 2.00 mm under a forming pressure of 6 bar. The filament winding process was simulated using CADWIND software (version 10.2), resulting in a three-layer winding scheme consisting of two helical layers (19.38° winding angle) and one hoop layer (89.14°). The calculated thickness of the composite winding layer was 0.375 mm, and the coverage rate reached 107%. Finite element analysis, conducted using Abaqus, revealed that stress concentrations occurred at the knuckle region connecting the dome and the cylindrical body. The vessel was predicted to fail at an internal pressure of 5.00 MPa, primarily due to fiber breakage initiated at the polar transition. Experimental hydrostatic burst tests validated the simulation, with the vessel exhibiting failure at an average pressure of 5.06 MPa, resulting in an error margin of only 1.2%. Comparative tests on vessels without adhesive sealing at the head showed early failure at 2.46 MPa, highlighting the importance of head sealing on vessel integrity. Scanning electron microscopy (SEM) analysis confirmed strong fiber–matrix adhesion and ductile fracture characteristics. The close agreement between the simulation and experimental results demonstrates the reliability of the proposed design methodology and validates the use of CADWIND and FEA in predicting the structural performance of composite pressure vessels.

## 1. Introduction

With the growing demand for high-performance and lightweight structural systems, composite pressure vessels have gained widespread application in aerospace, energy storage, transportation, and fire safety. Compared to traditional metal-lined vessels, composite pressure vessels offer superior specific strength, corrosion resistance, and design flexibility. In particular, for hydrogen storage applications, composite vessels have become a key enabling technology for safe and efficient energy containment. Glass fiber-reinforced composites are especially suitable for low-pressure vessels due to their cost-effectiveness, ease of processing, and favorable mechanical performance.

Typically, composite pressure vessels consist of three components: a liner, fiber winding layers, and a resin matrix. The liner, commonly made of high-density polyethylene (HDPE), provides gas tightness and dimensional stability under internal pressure. The fiber winding layers bear the majority of the mechanical load, and the resin matrix serves to bind the fibers, transfer stress, and enhance structural integrity. Filament winding technology, which allows for the precise control of fiber placement, angle, and tension, is crucial for achieving high-performance and structurally efficient composite shells.

In recent years, numerical simulation techniques have become indispensable in the design and optimization of composite pressure vessels. Finite element analysis (FEA) is widely used to predict stress distributions, deformation behaviors, and failure modes under operational conditions. For example, Saharudin conducted a comparative numerical study on Type V hydrogen pressure vessels made of Kevlar/Epoxy, Basalt/Epoxy, E-Glass/Epoxy, and Carbon T-700/Epoxy [1]. The study reported burst pressures ranging from 81.33 MPa to 140 MPa, with the carbon fiber-reinforced vessel exhibiting the highest strength, highlighting the importance of material selection in pressure vessel design. Moreover, the fiber winding angle is a critical parameter influencing mechanical performance. Wang demonstrated via FEA that optimized winding angles significantly improve the structural strength and burst resistance of composite vessels [2]. This underscores the importance of integrating simulation tools with process parameter design. Multi-scale modeling approaches have also been developed to capture both microscale fiber architecture and macroscale structural responses. Recent studies using multi-scale finite element frameworks have accurately predicted burst pressures and failure locations in filament–wound structures. Such modeling techniques provide enhanced predictive capability for complex composite geometries [3].

While high-pressure Type IV and Type V composite vessels have been extensively studied, systematic research on low-pressure glass fiber-reinforced vessels remains limited, particularly in terms of integrated studies covering liner formation, winding simulation, structural analysis, and experimental validation. Therefore, this work focuses on the comprehensive design and evaluation of a low-pressure glass fiber composite vessel. This study includes the HDPE liner design and blow molding process, filament winding pattern generation using CADWIND, structural performance prediction via FEA in Abaqus, and final validation through hydrostatic burst testing. The goal is to establish a validated design methodology for low-pressure composite pressure vessels, offering a practical reference for future engineering applications.

## 2. Finite Element Analysis of Blow Molding of Pressure Vessel

### 2.1. Structure Design of High-Density Polyethylene Lining

Due to the storage function of the inner liner of the composite pressure vessel, which is used to store gas or liquid, it is necessary to have a good sealing property in the manufacturing of the mandlet to prevent leakage [4], and high-density polyethylene has the characteristics of high density, good air tightness and sealing, and good compatibility with the medium. In recent years [5], it has gradually begun to be used in the Type IV pressure vessel of composite winding. Because the manufacture is a low-pressure vessel, the core mold should have a certain thickness for carrying a certain pressure, and in the winding molding, it also needs to have a certain stiffness to prevent deformation during winding and affect the winding result.

The fire extinguisher shell features a cylindrical rotational structure, consisting of a central cylindrical body connected to elliptical heads with pole holes at both ends. Glass fiber-reinforced composite material is wound around the inner liner to form the outer shell. Although this configuration accurately represents the actual structural conditions, its complex geometry significantly increases the difficulty of modeling and the computational cost in finite element analysis. To improve modeling efficiency, reduce computational resource consumption, and focus on the key structural regions that most significantly influence mechanical performance, an appropriate structural simplification of the liner model is adopted in this study. The simplified model retains the essential load-bearing characteristics while omitting geometrical details with minimal impact on the analysis results. The final simplified geometry is illustrated in Figure 1.

### 2.2. Lining Forming Process

Because the inner lining of the mandrel features a circular symmetric structure, it is well suited for blow molding and offers economic advantages. The influence of the blow molding process on the quality and performance of the final product has been investigated. The quality of a blow-molded part primarily depends on key processing parameters, such as blow-up ratio, mold temperature, blow pressure and speed, and material shrinkage [6]. The blow-up ratio is defined as the ratio of the maximum diameter of the plastic part to that of the mold. To ensure smooth formation of the preform, an appropriate blow-up ratio is essential. According to McKeen’s study [7], a blow-up ratio (BUR) ranging from 2:1 to 4:1 is considered optimal for the blow molding process. In this study, the blow-up ratio is 2:1. With this blow-up ratio, a relatively uniform wall thickness can be achieved while also taking material cost into consideration [8]. Accordingly, the final billet is designed with a radius of 42.2 mm. Blow molding is a tube embryo formed by heating and melting plastic particles through the extruder and then extrusion through the screw through the mold. When the temperature of the embryo is high, the fluidity of the embryo is relatively good, which can easily lead to the plastic blow deformation and cut the product outline clearly, but the ability to maintain the shape is poor, the pressure holding time and cooling time are long, and the embryo transfer process can easily be damaged by external influences, affecting the production cycle of the product. On the contrary, when the embryo temperature is low, its molding ability will be relatively poor, there will be some stress concentration, residual stress in the molded parts will affect the strength of the product, but it is also easy to produce warping deformation, affecting the shape of the product [9,10]. Blow molding pressure is the pressure of compressed air into the cavity, and blow molding speed means the flow rate of compressed air into the cavity. It is obviously necessary to choose different blow pressures and blow speeds according to different volumes and wall thickness products [11]. Generally speaking, in actual production, a pressure of 2–7 bar is usually selected [12]. For thin-walled containers with large volumes or products featuring delicate surface patterns, the blow molding pressure can be appropriately increased to ensure complete cavity filling and pattern replication [13]. At the same time, a relatively high blow molding speed, i.e., a high flow rate of compressed air, is generally preferred. A higher airflow rate allows the cavity to be filled more rapidly, enabling the parison to better conform to the mold’s inner surface. This helps minimize the influence of gravity on wall thickness and promotes the formation of plastic parts with uniform wall thickness and smooth surfaces [14]. However, if the blowing speed is excessively high and exceeds the mechanical strength of the parison, it may cause the parison to rupture, leading to defective or wasted products.

The material selected for the product is high-density polyethylene (HDPE). The HDPE used in this study was in the form of injection-grade pellets (Model: 5502), supplied by Yanshan Petrochemical Co., Ltd. (Beijing, China). Considering the product’s relatively small volume and moderate wall thickness, a preliminary internal blowing pressure of 6 bar is proposed. This value is determined based on a combination of empirical data and practical requirements for HDPE blow molding. According to the literature and industrial practice, the typical blowing pressure for small- to medium-sized HDPE containers ranges from 4 to 8 bar [15], depending on the wall thickness, mold geometry, and desired surface finish. A pressure of 6 bar is chosen to ensure sufficient expansion of the parison and full conformity to the mold cavity, which is particularly important for achieving dimensional accuracy and surface quality. At the same time, this pressure remains below the critical limit that may cause thinning or rupture of the HDPE parison, thus ensuring safety and process stability. The shrinkage rate of blow-molded parts is the ratio of the size of the product reduced after the cooling of blow-molded parts to the size of the molding due to the influence of high temperature, thermal expansion and cold contraction. Under normal circumstances, the shrinkage rate of the blow-molded container has little effect on it. At the same time, hollow blow molding is the use of compressed air to blow the billet to fit the surface of the cavity molding. The size of plastic parts is related to air pressure, billet temperature, molding cycle, geometry and so on. The material exhibits good processability, chemical resistance, and mechanical strength, making it suitable for blow molding applications. The typical shrinkage rate of HDPE ranges from 1.5% to 3.5% [16], and a value of 2% was adopted in this work for dimensional compensation during product design. The final product obtained via blow molding is shown in Figure 2.

### 2.3. Finite Element Analysis of Lining Forming Process

The finite element model was constructed using NX UG 12.0 3D modeling software, developed by Siemens (Munich, Germany). The software used in this study was a licensed version authorized for our institution. In this software, Boolean operations were applied to create two cavities, thereby dividing the mold, as illustrated in Figure 2. Reposition the two cavities so that they are 100 mm apart, and construct a model of an embryo with a radius of 42.2 mm at the center. The surfaces of the extracted embryo and two cavities are shown in Figure 3. Because this simulation simulates the actual blowing process, only part of the formed surface is extracted to simplify the model.

In the meshing interface, the parison is defined as a fluid material, and its bottom and top edges are set as boundary conditions. The expansion pressure is applied between these two defined boundaries. Mold closing is achieved when the two cavities move toward each other along the *X*-axis. The motion of the cavities is controlled by velocity and time parameters. Specifically, the velocity in the X direction (Vx) is set to 0.5 m/s. The relationship between cavity velocity (Vx) and time (*t*) is described by the slope function (Equation (1)). According to this function, the cavities move at a constant speed of 0.5 m/s within the time interval *t* = 0–0.1 s. Once *t* ≥ 0.10001 s, the cavities stop moving. The direction of movement is controlled by the sign of the velocity value. Since the initial distance between the two cavities is set to 100 mm, they can fully close the mold within 0.1 s under the given motion function. The velocities in the Y and Z directions are set to zero, as there is no movement along these axes.(1)$$f\left(t\right)=f\left[\left(a,b\right),\left(c,d\right)\right]$$

By substituting the values *a* = 0.1, *b* = 1, *c* = 0.10001, *d* = 0, the curve of the ramp function is shown in Figure 4.

First, set the properties of the material. The material of the embryo type is high-density polyethylene. The data of dynamic viscosity μ = 10^4^ Pa·s and density ρ = 900 kg/m^3^ are used to define the high-density polyethylene in the finite element method. The influence of inertia is considered to simulate the relatively thick bottom embryo in actual production. The mold thickness is 3 mm, and the two cavities are defined as adiabatic molds.

The blowing pressure is applied directly to the inner surface of the parison. Accordingly, the predefined blowing pressure of P = 60,000 Pa is directly input into the simulation. The time-dependent variation in the blowing pressure is defined using the slope function (Equation (1)). The parameters of the function are assigned as follows: a = 0.1, b = 0, c = 0.4, and d = 1. The corresponding function curve illustrating the pressure–time relationship is shown in Figure 4. The function image indicates that in the time interval of 0~0.1 s, compressed air is not injected. During this time, the mold closing action of the two cavities is completed and, then, at 0.1~0.4 s, slowly reaches the set pressure of compressed gas.

The calculated wall thickness distribution is shown in Figure 5. As illustrated, when using a mold with a radius of 42.2 mm, a mold tube wall thickness of 3 mm, and a blowing pressure of 6 bar, the resulting bottle wall thickness ranges from 1.39 mm to 2.00 mm. The fact that the designed inner liner has a nominal wall thickness of 1.5 mm suggests that the selected process parameters are appropriate and effective.

## 3. Design and Simulation Analysis of Winding Process

### 3.1. Selection and Determination of Fiber Materials

Compared with other fibers, glass fiber has excellent corrosion resistance, convenient manufacturing, low price, and is more and more widely used in the civil field. The E6 386T is a straight roving without twist. It is coated with an infiltrating agent that contains silyl. When it is used, it is mainly suitable for reinforced unsaturated polyester resins and can also be used for vinyl resins, but it is mostly used for epoxy resins. It can be used for winding molding process products, pultrusion molding products, and the weaving process. 386T glass fiber can be used in the manufacture of pressure vessels, the molding and manufacturing of glass steel pipes, boats, large containers for industrial use, etc. E6 386T glass fiber has low storage conditions and is easy to store. In the absence of special requirements, glass fiber products should be stored in a dry, cool place to prevent moisture.

E6 386T glass fiber, supplied by China Jushi Co., Ltd. (Tongxiang, China), was selected as the reinforcement material for this study due to its cost-effectiveness, high mechanical strength, and excellent corrosion resistance. As a widely used direct roving product in composite applications, E6 386T is particularly suitable for pressure vessel manufacturing. In this work, it was employed to investigate the filament winding process of composite fire extinguisher cylinders.

### 3.2. Selection of Winding Forming Process

During the filament winding process, the movement of the fiber is continuous, and its trajectory is defined by the geometry of the pressure vessel (or other mandrel structure) and the motion of the winding nozzle [17]. For typical pressure vessels, the filament winding process can generally be classified into three types: hoop winding, helical (spiral) winding, and polar (longitudinal) winding. However, both hoop and polar winding can be considered as special cases of helical winding, where the winding angle approaches 90° or 0°, respectively. The distinction among these processes primarily arises from variations in the fiber winding angle relative to the vessel axis.

Among the three winding methods used in the filament winding process, each has its own advantages and limitations to varying degrees [18]. Among them, the spiral winding method is the most widely applied, as it is suitable not only for products with regular geometries but also for those with irregular shapes. However, this method also involves the most complex parameter design and process control in the winding process [19]. Despite the challenges, spiral winding typically yields the best overall performance among the available techniques. Circumferential winding offers a simpler design, but it becomes unsuitable when dealing with irregular shapes or products with pronounced surface protrusions. Longitudinal winding is primarily used for ellipsoidal products. Since products vary in shape, the optimal winding method must be selected based on the specific requirements of the forming process to ensure the desired mechanical performance [20]. Spiral and circumferential winding methods can be effectively combined to enhance the structural integrity of the final product. In this study, the adopted approach begins with an initial layer of spiral winding, followed immediately by a second spiral winding layer. The process then transitions to the opposite end of the mandrel, where spiral winding resumes [21].

The linear shape of the spiral winding is related to the position and number of tangential points; that is, the specific arrangement of the fiber is related to the number of turns passed in the middle of the second tangent of the linear, that is, the number of tangential points. The specific arrangement of fibers is studied; that is, the number of cut points and the distribution of cut points at the head of the fibers are studied [22].

Generally speaking, the number of tangential points can be divided into single tangential points and multi-tangential points. In the current study, most of the multi-tangential points are adopted; that is, there are multiple tangential points when the fiber is winding and forming, and the second circle of the fiber is not tangential to the fiber of the previous circle. After a back and forth, the fiber of the last place is tangential. When the number of cut points is more than or equal to three, the cut points will be ordered in a different way. As mentioned earlier, the three-cut points have two ordering orders, the four-cut points have two ordering orders, and the five-cut points have four ordering orders. Figure 6 shows a line diagram with five tangential points.

When the number of tangential points is *n*, the winding law of this line is as follows [23]:(2)$$\theta_{n-K}=\left(\frac{K}{n}+N\right)\times 360^{\circ }\pm \frac{\Delta \theta }{n}$$
where *K* is the *K* initial tangential points adjacent to each other in the winding law of the *n* tangential points. In the polar hole, the circumference arrangement order is different, or when the guide wire head moves back and forth once, the center angle law that the mandrel must turn is different, where *K* is a positive integer (*K* = 1, 2, …, *n* − 1). *K* values require that *K*/*n* be the simplest fraction.

Once the structural configuration of the composite pressure vessel is determined, the winding angles are subsequently calculated. Given that the front and rear polar holes of the vessel are identical in size, the helical winding angle is determined to be 19.38°, and the hoop (circumferential) winding angle is 89.14°. The thickness of a single winding layer is 0.1282 mm. These parameters were calculated based on Andrews’ equation, which considers the geometry of the mandrel and the fiber winding path to ensure an optimal stress distribution and structural performance [24].

### 3.3. Simulation Analysis of CADWIND Winding Molding

Fiber winding process design simulation software CADWIND (version 10.2) is a professional software with the functions of winding pattern design, winding program calculation and product assistant design. The main function of the software is divided into five parts: mandrel creation, winding line design, winding program programming, machine tool motion simulation and product design. The software module includes geometric mandrel modeling, winding profile calculation (dotting point trajectory calculation), winding program calculation (NC numerical control code), machine tool parameter setting, machine tool dynamic simulation of fiber winding process, winding layer finite element data output and laminate strength calculation, etc. It is a professional simulation software integrating CAD/CAM/CAE in the process of fiber winding design. The software can not only be applied to linear design and the winding program of axisymmetric core molds such as straight cylinder, pressure vessel, bottle and sphere. It can also be applied to the linear design and winding programming of non-circular-section straight pipes, such as rectangular section and oval section, circular section, rectangular section and oval section bending pipe, and T-shaped parts and other non-axisymmetric mandrels.

From the aspect of geometric mandrel modeling, the software provides automatic modeling and external CAD mandrel input functions. From the aspect of winding design, it provides some typical winding design functions, such as circular winding, spiral winding, longitudinal plane winding, non-geodesic winding, and the function of calculating result dialogue window and calculating result cloud image analysis.

CADWIND software also developed the data interface of structural strength finite element software, output the real winding process of the mandrel geometric mesh element, laminate thickness, winding angle size. Creating a geometric mandrel with CADWIND simulation software, the first step in the operation process is to create a computer model of the geometric mandrel of the wound part. Software creates mandrels in two ways. The first method uses a mandrel generator to quickly and easily create computational mandrel models of common geometries. The second method, for some complex-shaped mandrel, can be input through DXF geometry file or stylistic format file.

In this study, the first method is used to construct the mandrel of pressure vessel winding. The mandrel parameters are shown in Table 1.

After establishing the mandrel through CADWIND, the mesh of the mandrel is divided through CADWIND by adjusting the mandrel data. The mandrel model is established by CADWIND, and the parameters are the parameters of the inner liner when the mold is designed, as shown in Figure 7.

After establishing the mandrel, set the material parameters, as shown in Table 2 and carry out the layup design of the winding layer of the mandrel. After the material parameters are set, a series of winding data can be obtained according to the set material parameters, and the corresponding model can be generated, see Table 3.

Combined with the design analysis of the linear, in order to avoid the overhead phenomenon of the fiber at the pole hole, the linear with fewer cutting points should be selected as far as possible, and the coverage of the winding layer can reach 100% or more. CADWIND winding simulation software was used to analyze the number of winding turns, the number of tangling points and the coverage degree, and the first set of data was selected. The number of tangling points was 5, the number of winding turns was 67, and the coverage was 107%. After determining the five-point winding pattern, on the basis of the five-point winding pattern, a single bundle of yarn is selected to analyze the arrangement law of the winding fiber on the inner liner, and then the finite element data are output through the CADWIND winding simulation software; the finite element software is analyzed. The winding parameters are shown in Table 3.

## 4. Simulation Analysis of Glass Fiber Composite Pressure Vessel

In the fiber pressure vessel, the continuous process of composite stress damage to the lower plate is more complicated: first, the transverse cracking of one layer leads to damage to the other layer. Only a few monolayer layers are based on their longitudinal fibers before the laminate finally fails, in which case it can be assumed that the flexible matrix modulus is zero, i.e., em = 0, so the essence of the grid theory is to assume that the stiffness of the matrix deteriorates to zero after transverse fracture. According to this assumption, the film force is completely borne by the fibers. Let the fiber stress and the fiber thickness be the circumferential film stress and the axial film stress, respectively, which can be expressed as Formulas (3) and (4), respectively [25].(3)$$N_{\theta }=\sigma_{f}t_{f}\sin^{2}\left(\alpha \right)$$(4)$$N_{\phi }=\sigma_{f}t_{f}\cos^{2}\left(\alpha \right)$$
where *α* represents the winding direction of the fiber and the angle of the central axis of rotation. Thus, we obtained(5)$$tg^{2}\alpha =N_{\theta }/N_{\phi }$$

The stress of the fiber can be described by Formula (6).(6)$$\sigma_{f}=N_{\phi }/t_{f}\cos^{2}\left(\alpha \right)=N_{\theta }/t_{f}\sin^{2}\left(\alpha \right)$$

### 4.1. Establishment of Finite Element Model

The finite element analysis (FEA) model was developed using CADWIND filament winding software in combination with Abaqus 2021 for numerical simulation. First, the winding mandrel was created in CADWIND, followed by meshing of the mandrel. The composite layers were generated through a non-geodesic helical winding process, including spiral winding, mandrel updating, hoop winding, another mandrel update, and a final spiral winding. The complete model was then exported in INP format and imported into Abaqus for finite element analysis [26]. All simulations were performed using a licensed version of Abaqus under an academic research license agreement.

(1)Element Selection and Mesh Division

There are various ways to classify finite elements. Shell elements, for instance, can be categorized based on thickness—into thin-shell and thick-shell elements—or by application, such as general-purpose shell elements or those designed for specific scenarios [20]. Elements can also be defined by their formulation, including conventional shell elements or continuum shell elements. In modeling fiber-reinforced composites, a variety of finite element types can be used [21], including solid elements, shell elements, and beam elements. Given the thin-walled nature of the composite winding layers and the primary interest in in-plane mechanical behavior, shell elements were selected for this study. Among the available shell elements, the S4R element—a four-node quadrilateral shell element with reduced integration and hourglass control—was chosen due to its high computational efficiency and accuracy. It also supports large deformation analysis and is well suited for orthotropic material modeling. To accurately capture the thickness variation at the structural head and avoid poor-quality distorted elements, the sweep meshing method was not adopted. Instead, a structured meshing strategy was applied based on the geometric features of the model. The geometry was first partitioned into distinct regions, followed by mesh control through the placement of key nodes to ensure uniform and smooth element distribution across the structure. After meshing, the model consisted of 5240 nodes and 5200 S4R shell elements, as shown in Figure 8, meeting the requirements for accuracy and stability in subsequent finite element analyses.

The quality of the mesh plays a critical role in the accuracy and convergence of finite element analysis. A well-constructed mesh contributes to reliable simulation results and reduces computational time. Therefore, after meshing, it is essential to perform a thorough quality assessment. In this study, the mesh quality was evaluated based on standard indicators, such as element skewness, aspect ratio, and Jacobian determinant. The results showed that the overall mesh quality met acceptable thresholds. However, in areas with complex geometry—such as sharp corners, transitions, or narrow features—a few distorted and highly skewed elements were identified. These low-quality elements were located through mesh diagnostics and subsequently refined or adjusted locally to eliminate potential issues that could compromise the accuracy and convergence of the simulation. Through this process, the mesh uniformity and overall calculation stability were significantly improved, thereby enhancing the accuracy of the numerical results.

(2)Material parameter setting

The anisotropy of fiber-reinforced composites makes the design of fiber-reinforced composite structures more difficult than that of traditional metal materials. At the same time, the designer can determine the direction of fiber reinforcement according to the main force direction of the structure, making the design more flexible.

Due to the morphological characteristics of the composite material itself, the crack propagation direction is inclined, and the crack direction may be caused not only by the load, geometric structure and boundary conditions but also by the microstructure of the material itself. The interface between the fiber and the resin is usually weaker than the surrounding material, and interface stripping is usually the first form of failure that occurs in fiber-reinforced composites. In addition, the coefficient of thermal expansion of the fiber and the resin, in the transverse and longitudinal state, is also different, and the residual thermal stress can also cause the generation of cracks.

The winding layer of the composite material used in this study is glass fiber composite material; the model is E6 386T, its mechanical properties are affected by the winding tension of the fiber, the winding tension is different, and the performance is different, but it is an orthogonal anisotropic E6 386T glass fiber composite material and has the typical mechanical properties of composite materials. The parameters are shown in Table 4.

Figure 9 shows that, in the finite element model, the direction of fiber winding is spiral winding, the winding angle is 19.38°, the winding angle is alternately winding, and the winding angle is 89.14°. The winding mode is the first layer; that is, the innermost layer is spiral winding, the second layer is circumferential winding, and the outermost layer is spiral winding. In the finite element model, there are six layers in the barrel, as shown in Figure 9a, and four layers at the head, as shown in Figure 9b.

(3)Boundary and load conditions

The whole structure of the composite pressure vessel is analyzed. Since the pressure vessel is a rotary body, it only needs to apply displacement constraints to one of the polar holes, add the boundary conditions of the constraints, and the rest of the free expansion. The inner liner of the composite pressure vessel is wrapped by the fiber full winding structure, and there is no relative sliding between the inner liner and the winding layer structure. The boundary fixation conditions are shown in Figure 10a. On the surface of the cylinder body, a uniform internal pressure is applied, as shown in Figure 10b.

### 4.2. Simulation Result Analysis

The failure mechanism of fiber-reinforced composites differs significantly from that of traditional metallic materials [27]. Unlike metals, which typically exhibit yielding and plastic deformation under tensile stress, most fiber-reinforced composites display a distinctly brittle failure behavior. This means that instead of undergoing large plastic deformation, composites tend to fracture abruptly once their strength limit is reached. However, at the microscopic scale, these materials will further appear in some damage forms, such as matrix cracking and fiber fracture, thus exhibiting a mechanism similar to yield, which is the energy absorption mechanism. Under external load, the composite laminates show the characteristics of gradual deterioration. When local damage occurs, the whole structure will not suddenly fail [28]. The failure process of fiber-reinforced composites is a gradual failure process, not a sudden failure process, and its damage mechanism and evolution process are quite complicated. At the initial stage of damage, there will be many tiny cracks in the matrix, and these tiny defects may not be shown at the macro level, but with the continuous accumulation of damage, the connection interface between the fiber and the matrix will be stripped; then, the fiber fracture will occur. Once the fiber is broken, there will be a redistribution of load. The mechanical properties of the composite materials will continue to deteriorate, and the carrying capacity of the material will be reduced, which is manifested in a reduction in the elastic modulus and Poisson’s ratio of the material at the macro level. When the pressure vessel is subjected to an internal pressure of 1.5 MPa, the resulting strain and stress distribution contour plots are shown in Figure 11a and Figure 11b, respectively.

It can be seen from the above figure that the winding layer of glass fiber composite material has greater stress and strain in the cylindrical part of the cylinder. It can be concluded from the simulation results that the stress–strain gap between the stress of the spiral wound layer and that of the circumferential wound layer is larger than that of the axial stress when the composite pressure vessel is subjected to uniform internal pressure, and the elastic modulus of the fiber in its own direction is the largest. Compared with the helical wound layer, the toroidal wound layer bears more tensile load, so the stress–strain in the toroidal wound layer is larger than that in the helical wound layer. At the place where the head and the barrel body are tangent, the stress is concentrated, which is caused by the gradual thickening of the head at the polar circle where the head and the barrel body are tangent. Because the head is an oval head, its structure is special. After the pressure vessel is subjected to uniform pressure, the stress it is subjected to is relatively low compared with the stress of the middle barrel part. At the same time, due to the winding mode of the glass fiber composite pressure vessel, the winding mode inside the cylinder is spiral winding, then circumferential winding, and then spiral winding is added after the end of the circumferential winding, while the winding mode at the head is only spiral winding, without fiber wrapping, two spiral winding, and a layer of circumferential winding is separated between the head and the barrel, resulting in discontinuity when the head and the barrel are wrapped. Thus, a concentration of stress occurs at the polar circle. Therefore, the analysis shows that the outer head of the winding layer is most vulnerable to damage. Under the internal pressure of 1.5 MPa applied at this time, the stress–strain value of the wrap layer head at the polar circle does not reach the damage value. The inner lining structure of the pressure vessel is high-density polyethylene. Under the action of internal pressure, the barrel body is wrapped with spiral wrap layer and circumferential wrap layer, while the seal has only a spiral wrap layer with small thickness, and the deformation degree at the wrap head is greater than that at the barrel. According to the finite element simulation analysis results, although the internal pressure of 1.5 MPa is easy to damage, it is not enough to make the container fail.

When a uniform internal pressure of 3 MPa is applied, the result is roughly the same as that of 1.5 MPa, which is not enough for the container to fail. When the pressure is continuously applied to 5 MPa, the container fails. The stress–strain cloud diagrams are shown in Figure 12.

According to the Tsai–Wu failure criterion, failure occurs when the failure index exceeds 1. The failure index is a dimensionless parameter, which evaluates the proximity of the stress state to the material’s failure envelope based on the anisotropic strength properties of the composite. Under an internal pressure of 5 MPa, the pressure vessel reaches failure conditions. Based on the combined analysis of the stress–strain distribution and the Tsai–Wu failure index, the failure is found to initiate at the knuckle region, i.e., the tangent point between the dome and the cylindrical body. The failure propagates from this critical region toward the barrel section. As the barrel wall expands outward, fiber breakage occurs, ultimately leading to structural failure. The failure mode and location are illustrated in Figure 13. This conclusion will be further verified through dedicated experimental investigations in the subsequent sections.

## 5. Experimental Verification Analysis

### 5.1. Test Scheme Design

The final performance of the composite pressure vessel is closely related to the winding method and the number of winding layers. According to the results of finite element simulation, it is determined that the three-layer winding method can meet the requirements. First of all, according to the finite element simulation results, the specific experimental scheme is designed. Due to the threaded connection at the head, considering the tightness of the container, the experimental scheme was first determined to verify the sealing effect at the head. Experimental scheme 1: The root winding mode is alternately spiral winding and circumferential winding, and the third layer is also spiral winding. According to the design results presented in Section 2, the total number of helical winding turns is 67, with a helical winding angle of 19.38°, a hoop winding angle of 89.14°, and a fiber band width of 5 mm. After the first layer of spiral winding is fully spread, the second layer of spiral winding begins through the circumferential transition. After completing the loop winding, the third layer of spiral winding is carried out again through the loop over-winding. When the winding is completed, the barrel body part has six layers and four layers at the head. Experimental scheme 2: Study the amount of winding fiber and glue, and find the best experimental results when the amount of fiber and glue is relatively small. The sealing head is coated with glue, and the winding method is the same as scheme 1. The number of spiral winding circles is divided into 62, 57, 52. In the two schemes, the spiral winding angle is 19.38°, the ring winding angle is 89.14°, and the line width is 5 mm, which are the same. The details are shown in Table 5.

### 5.2. Winding Forming and Curing

E6 386T glass fiber and the TECHSTORM 2402E/2402H epoxy resin system were used as composite layers to reinforce the glass fiber pressure vessel. In the experimental process, the TECHSTORM 2402E epoxy resin and TECHSTORM 2402H curing agent, both manufactured by Dawn Tianhe Materials Technology Co., Ltd. (Shanghai, China), were employed and mixed at a mass ratio of 100:80, as recommended by the manufacturer. In order to ensure uniform mixing, professional mixing and dispersing equipment was used. TECHSTORM 2402E/2402H is an epoxy system material for fiber glass winding applications. The product is 100% free of volatile components and consists of two parts: epoxy TECHSTORM 2402E and curing agent TECHSTORM 2402H. The epoxy system material can complete the curing reaction at a low temperature through the unique modification technology, which can meet the temperature control requirements in the production process of the plastic inner pressure vessel. In addition, the material of the system has good adhesion to the fiber, high mechanical strength and excellent heat resistance.

The glass fiber tows were fixed onto the rolling shaft of the tension control cabinet of the four-axis CNC filament winding machine (model: ZW-480), manufactured by Zhejiang WNJ Spring Machinery Co., Ltd. (Shengzhou, China). The TECHSTORM 2402E epoxy resin and 2402H curing agent were mixed at a mass ratio of 100:80, thoroughly stirred, and then poured into the resin bath. Prior to winding, the plastic liner was cleaned using acetone to remove surface contaminants. It was then connected to the spindle of the CNC winding machine via a shaft and fixed onto the triangular chuck, ensuring proper alignment and stability for the subsequent winding process.

First of all, debug the equipment, before the start of winding, including the four-dimensional CNC winding molding equipment for program design and debugging; set the zero starting point of winding molding, and idling a circle, study whether the trajectory of the nozzle in line with the design law of the program; if yes, start the formal winding; if not, adjust the zero starting point of winding. This can effectively avoid the phenomenon of slipping yarn when really winding, improve the success rate of the experiment, and reduce the waste of unnecessary experimental consumables.

When the motion track of the wire nozzle and the zero starting point of the winding are adjusted, the glass fiber is wound on the connecting shaft of the drive shaft through the impregnating system, and one end of the fiber is fixed on the drive shaft through the rotation of the spindle.

During the filament winding process, the resin impregnation system is a critical component where fibers are impregnated with resin, ensuring proper wetting and resin content control. This system typically comprises a resin bath and a set of rollers or doctor blades. In this experiment, a top-feed impregnation method was employed, where fibers are impregnated from above during the winding process. The setup is illustrated in Figure 14. The resin impregnation system used in this study was manufactured by Nantong Xiyuan Electromechanical Equipment Co., Ltd. (Nantong, China).

The curing method of composite pressure vessels varies with different resins [18]. The curing process is internal curing and external curing. One of the most used is external heating curing. External curing means that the pressure vessel is placed in the curing furnace, which is heated according to the set curing temperature course to complete the curing process of the composite material. In the external curing process, the composite material is heated layer by layer from the outside to the inside and the curing reaction occurs, resulting in the inner bubble and the gas generated during curing failing to be discharged, thus affecting the curing quality of the composite material. At the same time, the winding and molding of composite materials are carried out separately in two independent devices, which obviously limits the molding efficiency, resulting in high production cost and low performance of composite pressure vessels. Since the fiber-reinforced resin is wound and formed on the inner lining of the pressure vessel, and its structure is hollow and the inner lining material is metal, the composite material can be heated by heating the metal inner lining and applying the metal heat transfer theory to heat up the composite material, so as to complete the curing and forming. Compared with external curing, internal curing has obvious advantages; that is, in the process of internal curing, internal curing begins to be heated, thus improving the quality and carrying capacity of the pressure vessel. In addition, fiber winding and composite material curing are completed in one piece of equipment, which reduces the loading and unloading time and improves the production efficiency [29]. However, since this study studies a four-type pressure vessel with non-metallic lining, the external curing method is adopted. The wound pressure vessel products are taken out of the winding equipment and placed in another curing tank for curing. In this study, the filament–wound composite products were cured using a DGF150-25BF curing chamber, manufactured by Shanghai Yiheng Scientific Instrument Co., Ltd. (Shanghai, China). To ensure optimal mechanical properties and interfacial bonding of the fiber–wound pressure vessel, a multi-stage thermal curing process was employed. The total curing duration was 8 h, consisting of two temperature stages: an initial holding stage at 75 °C for 2 h, followed by a final curing stage at 90 °C for 6 h. The temperature was increased gradually at a controlled heating rate of approximately 2 °C/min to avoid internal thermal stress and bubble formation due to the rapid volatilization of residual solvents. To improve resin distribution and avoid localized accumulation or dry spots, a rotational curing technique was adopted. During the curing process, the pressure vessel was mounted on a rotating turntable and kept in continuous rotation. This ensured that the uncured resin remained evenly distributed across the surface under the action of gravity and capillary forces, particularly in polar and transition regions. To minimize resin splashing or fiber misalignment caused by centrifugal force, the rotation speed was kept low (typically 3–5 rpm), sufficient to maintain flow uniformity without disturbing fiber orientation. In addition, rotation helped ensure uniform thermal exposure and reduce the risk of warping or deformation. This combined approach of stepwise heating and rotational curing effectively enhanced the dimensional stability and surface quality of the final composite pressure vessel. A schematic of the curing setup is shown in Figure 15.

### 5.3. Blasting Test

Since the winding angle of the pressure vessel, the thickness of the winding layer fiber and the winding mode are all simulated by CADWIND fiber winding molding simulation software, the failure strength is calculated by the finite element simulation analysis software through the parameters and properties of the given material, the size of the load and the setting of the boundary conditions [30]. In order to verify whether the failure analysis method of the composite pressure vessel and the linear design method during winding molding are correct, the winding molding product is solidified, and the hydraulic blasting experiment is carried out to verify the accuracy of the design [9].

During the burst test, the pressure vessel was first completely filled with water to eliminate the presence of residual air, which could otherwise introduce experimental errors or pose safety risks due to gas compressibility during pressurization. Subsequently, hydraulic pressure was applied using a WH-50 high-pressure pump (manufactured by Jiangsu Wohua Testing Equipment Co., Ltd. Changzhou, China), with a maximum output capacity of 70 MPa. The pressurization was performed in a stepwise incremental manner until the vessel either ruptured or exhibited significant plastic deformation. The maximum pressure recorded prior to failure was defined as the burst pressure. Real-time pressure monitoring was achieved using a digital pressure sensor (model: PSM-200, manufactured by Shanghai Zhaohui Pressure Instrument Co., Ltd., Shanghai, China), with an accuracy of ±0.25% FS and a sampling rate of 10 Hz. To ensure data reliability, no fewer than three burst tests were conducted for each vessel specification. The average burst pressure and standard deviation were calculated to assess consistency and repeatability. The blasting experiment is shown in Figure 16.

### 5.4. Analysis of Experimental Results

The comparative experimental results derived from Experiment Plan 1 and Experiment Plan 2 are comprehensively illustrated in Figure 17. These results reflect the influence of different process parameters—specifically, the head sealing condition and the variation in the number of helical winding turns—on the overall mechanical performance and burst strength of the composite pressure vessel. In Experiment Plan 1, specimens featured adhesive sealing at the head and employed a consistent three-layer winding scheme with a total of 67 helical turns. In contrast, Experiment Plan 2 introduced variations in fiber content and resin usage, with a focus on evaluating vessels both with and without head adhesive sealing and with reduced helical winding turns (ranging from 67 to 52). The plotted data provide critical insight into the effectiveness of the sealing method and fiber layer distribution on structural integrity. Notably, the burst pressure of vessels without adhesive head sealing decreased significantly, confirming the pivotal role of proper sealing in preventing premature failure at the threaded joints. These experimental findings serve to validate the finite element simulation results and reinforce the importance of meticulous process control in the manufacturing of composite pressure vessels.

To further investigate the failure mechanisms of the glass fiber-reinforced epoxy composite pressure vessel, the macroscopic fracture morphology after burst failure was first examined, as shown in Figure 18. It can be observed that the fracture typically initiates at the junction between the dome and the cylindrical section, and then propagates axially along the vessel body. Subsequently, fracture surfaces were examined using scanning electron microscopy (SEM), as shown in Figure 19. Figure 19a,b present overviews at 300× magnification, while Figure 19c,d provide high-resolution views at 800×. The SEM analysis reveals a predominantly ductile fracture mode of the fibers, with smooth fracture surfaces and intact resin bridges between adjacent fibers. No significant interfacial debonding is observed, indicating strong fiber–matrix adhesion. In Figure 19d, the fiber bundles are clearly embedded in the matrix with cohesive interfacial contact. The fracture morphology exhibits a mixed “pull-out plus shear” failure mode, suggesting that the fibers sustained significant tensile loads before rupture, consistent with a synergistic failure process. Microcracking in the resin phase is also observed, characterized by tortuous, crack-propagation patterns and “turtle shell” textures, indicating that the matrix failed in a brittle manner before the fiber fracture. These microcracks appear near high-stress regions predicted by the finite element model, confirming that stress concentrations served as initiation sites for progressive damage. Furthermore, slight plastic deformation of some fibers is evident, implying a certain degree of energy absorption prior to catastrophic failure. This observation aligns with the macroscopic mechanical behavior and suggests that the resin–fiber interface effectively delayed crack propagation. Overall, the SEM analysis confirms the multi-stage, synergistic nature of damage in the composite material, beginning with matrix cracking, followed by limited interfacial separation, and culminating in fiber fracture. These findings are consistent with the simulation results and provide valuable microstructural insight into the material’s failure mechanism. Moreover, the strong interfacial bonding observed validates the effectiveness of the winding and resin impregnation process and offers guidance for future interface optimization in composite pressure vessel design.

The experimental data from scheme 1 were processed and analyzed. The data at the glue-coated section of the vessel head indicate that the experimental results align well with the simulation analysis. The pressure vessel fails at approximately 5 MPa. Based on the experimental outcomes, the actual burst pressure is around 5.06 MPa, representing a deviation of about 1.2% from the simulated failure pressure of 5 MPa. While finite element simulation typically yields more accurate results, uncertainties in the actual amount of adhesive applied during manufacturing can introduce some deviations. Nevertheless, the experimental error remains within an acceptable range, demonstrating that the designed pressure vessel is reliable and does not rupture prematurely.

According to the analysis of the experimental results of the uncoated group, the bursting pressure is 2.46 MPa, which indicates that the sealing property of the head has a great influence on the final performance of the product. When it is near 2.46 MPa, the rupture is caused by the poor sealing effect at the head. The head and the mouth of the pressure vessel are connected by thread, and no glue is applied when connecting. Water overflows from the threaded combination at the head, causing the fiber layer to break.

## 6. Conclusions

This study presents a comprehensive investigation into the design, simulation, and validation of a low-pressure composite pressure vessel reinforced with glass fibers. The research spans from HDPE liner design and blow molding, filament winding process simulation and finite element analysis to experimental verification. The main conclusions are as follows:(1)Feasibility of liner design and forming process

An HDPE liner with a wall thickness of 1.5 mm was designed and successfully fabricated using a blow-up ratio of 2:1 under a blowing pressure of 6 bar, resulting in a wall thickness distribution between 1.39 mm and 2.00 mm. Finite element simulations closely matched the actual molding results, confirming the reliability of the selected process parameters.

(2)Accurate winding process design with high coverage

Using CADWIND software, a three-layer filament winding scheme was established with a helical angle of 19.38° and a hoop angle of 89.14°, achieving a coverage rate of 107% and a total fiber layer thickness of 0.375 mm. Compared with Wang who recommended ±20° as optimal winding angles, the parameters used in this study also produced effective load distribution for low-pressure applications, demonstrating good adaptability of the winding scheme [2].

(3)Excellent agreement between finite element prediction and burst testing

The finite element model predicted failure initiation at the knuckle region under an internal pressure of 5.0 MPa. The experimental burst pressure averaged 5.06 MPa, with a deviation of only 1.2%, verifying the model’s accuracy.

(4)Good interfacial bonding and synergistic failure mechanism

SEM images revealed smooth fiber surfaces with no visible interfacial debonding, indicating strong adhesion between fibers and epoxy resin. The observed fracture mode was a combination of matrix cracking and fiber rupture, consistent with findings by Zhang, confirming the compatibility and effectiveness of the E6 386T fiber and TECHSTORM 2402E/2402H resin system [3].

(5)Significant impact of sealing on vessel performance

Specimens without adhesive sealing at the head exhibited a significantly reduced average burst pressure of 2.46 MPa, with failure localized at the threaded joint. This highlights the critical role of proper sealing in ensuring the structural integrity of composite pressure vessels.

In summary, this work establishes a reliable methodology for the integrated design, simulation, and experimental validation of glass fiber-reinforced composite pressure vessels. The findings provide valuable data and practical insights for the engineering application and optimization of similar composite structures.

## Figures and Tables

**Figure 1 materials-18-02485-f001:**
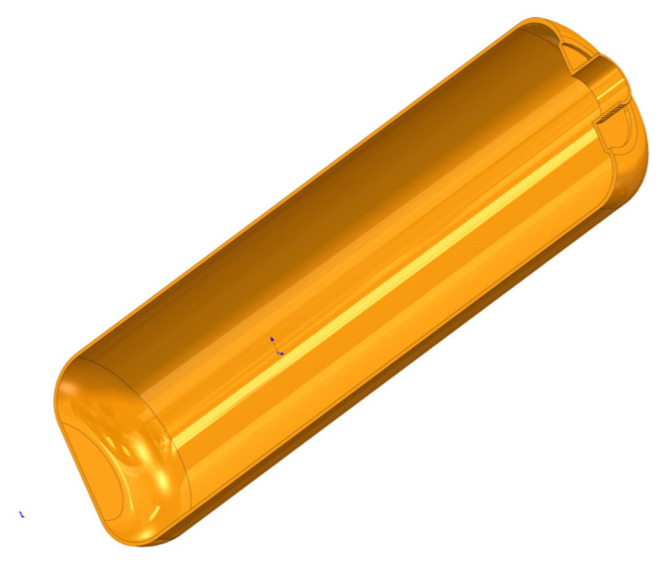
Lining volume 3D model.

**Figure 2 materials-18-02485-f002:**
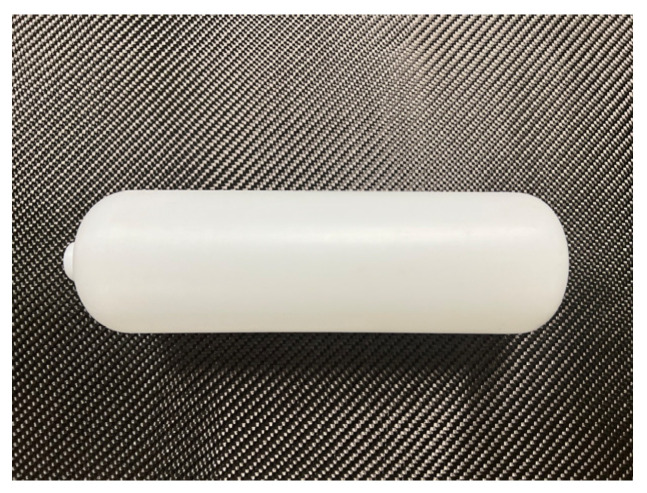
Lining volumetric molding products.

**Figure 3 materials-18-02485-f003:**
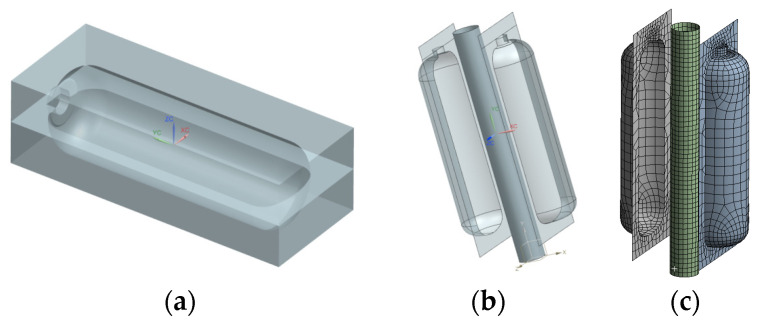
Finite element analysis of the liner forming process: (**a**) 3D assembly view. (**b**) Exploded view. (**c**) Meshing view.

**Figure 4 materials-18-02485-f004:**
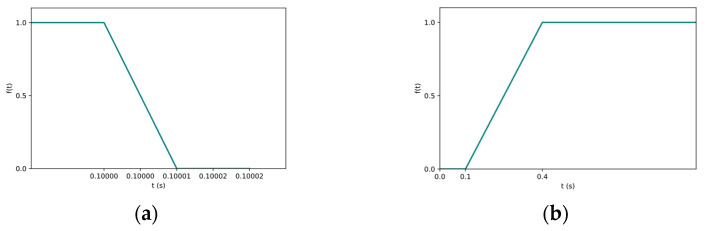
Boundary condition: (**a**) the curve of the relationship between speed and time; (**b**) the function relation curve of blowing pressure and time.

**Figure 5 materials-18-02485-f005:**
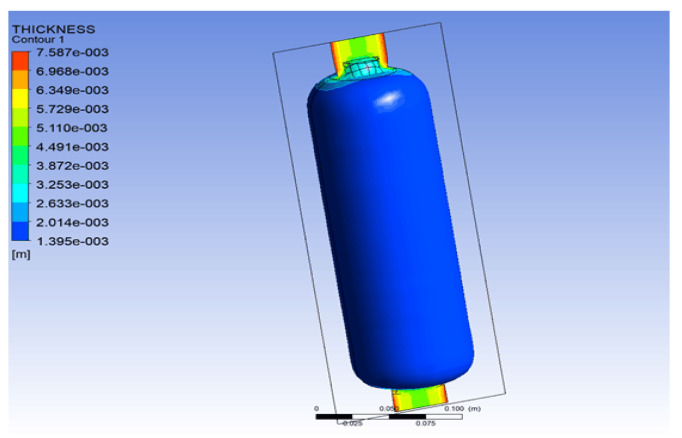
Wall thickness distribution diagram of blow-molded inner liner.

**Figure 6 materials-18-02485-f006:**
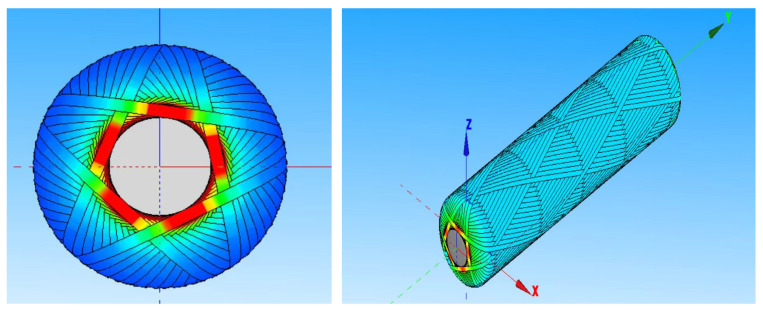
Five-tangent line diagram. The first image shows a cross-sectional view of the fiber winding structure with stress distribution, where colors represent different stress levels (red: high stress, green: moderate stress, blue: low stress). The second image displays the overall winding pattern along the cylindrical surface in three-dimensional space. The colored lines/arrows indicate fiber winding paths and orientations along the X, Y, and Z axes.

**Figure 7 materials-18-02485-f007:**
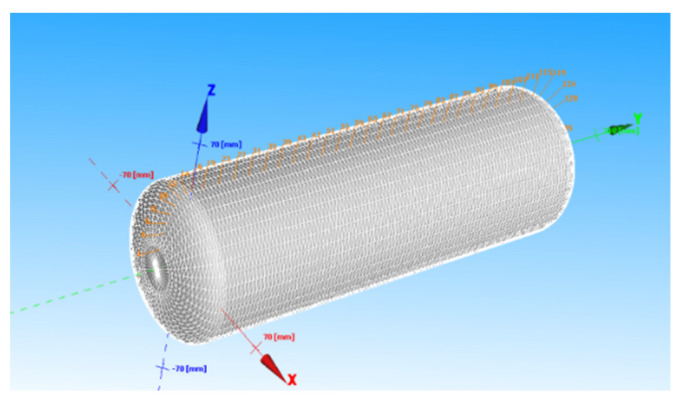
Mandrel grid.

**Figure 8 materials-18-02485-f008:**
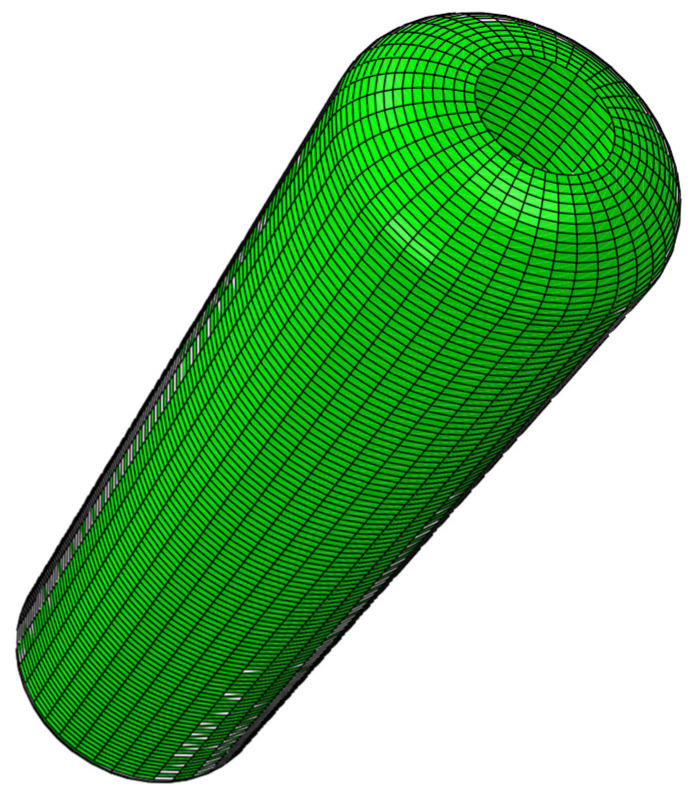
Meshing diagram of the composite pressure vessel for finite element analysis.

**Figure 9 materials-18-02485-f009:**
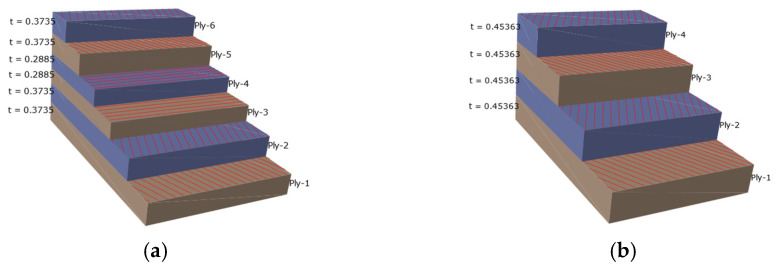
Composite paving setup: (**a**) the tube is laid; (**b**) cover the head.

**Figure 10 materials-18-02485-f010:**
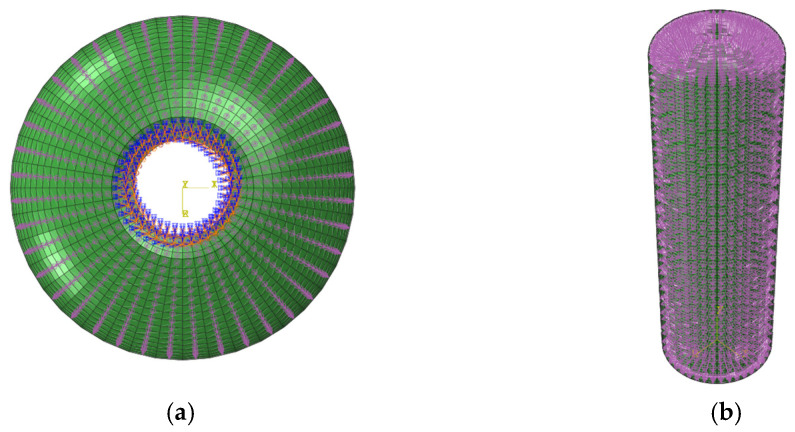
Boundary and load conditions: (**a**) boundary condition; (**b**) uniform internal pressure.

**Figure 11 materials-18-02485-f011:**
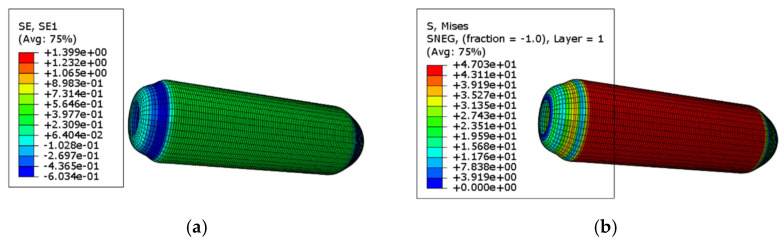
1.5 MPa stress–strain cloud diagram: (**a**) stress cloud diagram (unit: MPa); (**b**) strain cloud diagram (unit: mm/mm).

**Figure 12 materials-18-02485-f012:**
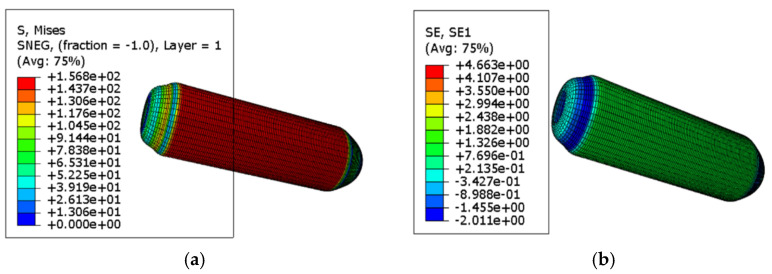
5 MPa stress–strain cloud diagram: (**a**) stress cloud diagram (unit: MPa); (**b**) strain cloud diagram (unit: mm/mm).

**Figure 13 materials-18-02485-f013:**
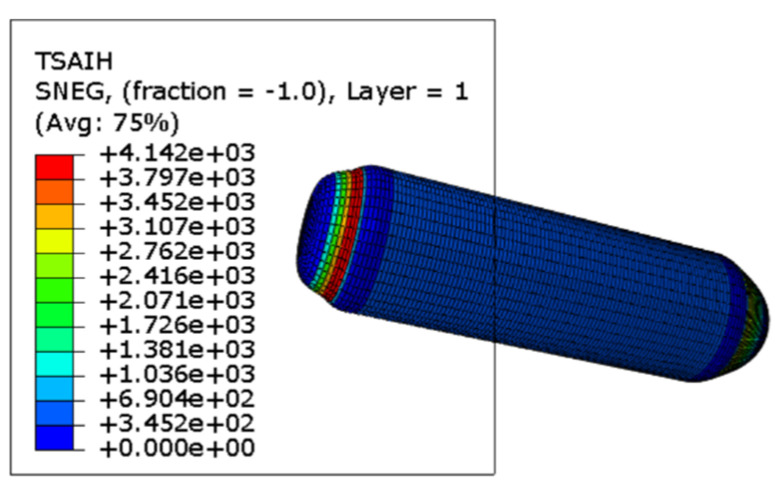
5 MPa Tsai–Wu failure diagram (unit: mm/mm).

**Figure 14 materials-18-02485-f014:**
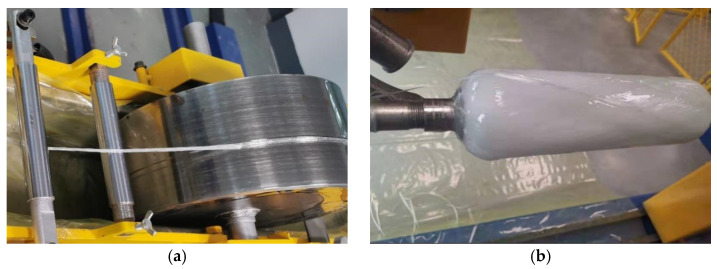
Fiber winding forming: (**a**) dipping system and method; (**b**) winding result.

**Figure 15 materials-18-02485-f015:**
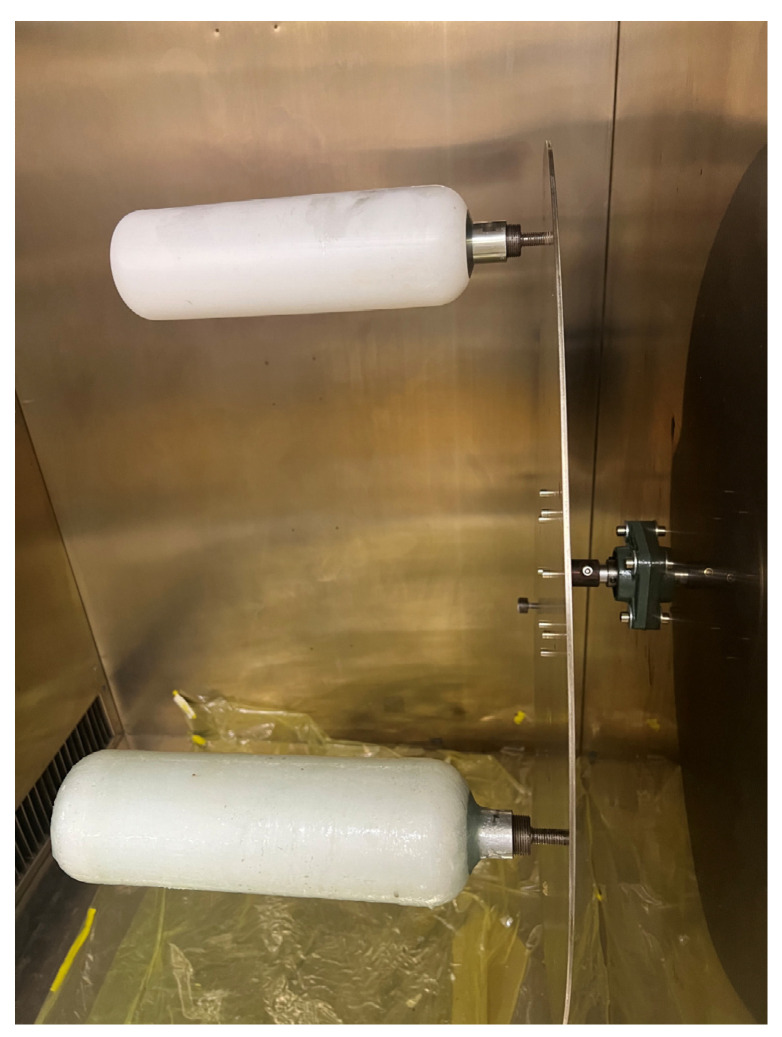
Curing box and curing method.

**Figure 16 materials-18-02485-f016:**
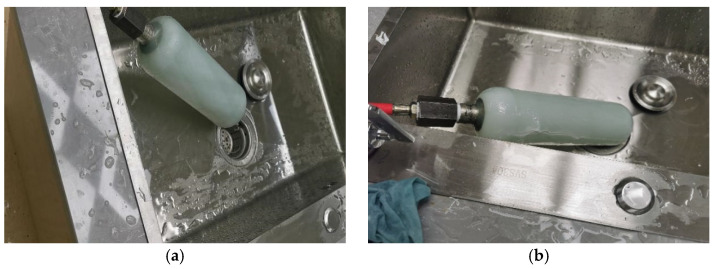
Pressure vessel bursting test diagram: (**a**) preburst condition; (**b**) post-blasting condition.

**Figure 17 materials-18-02485-f017:**
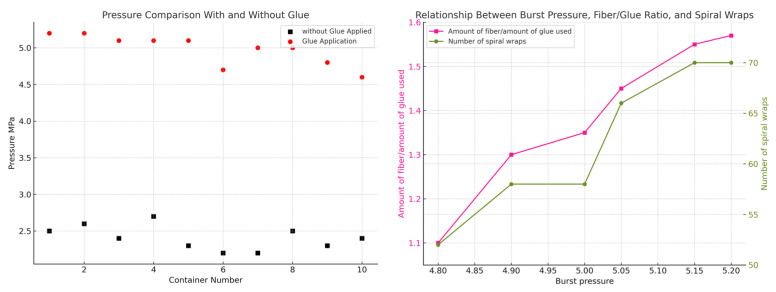
Comparison of data of two experimental schemes.

**Figure 18 materials-18-02485-f018:**
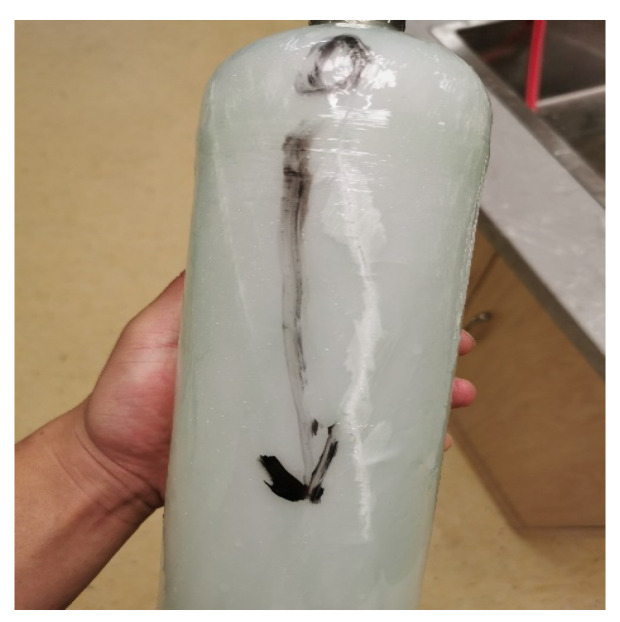
Fracture line of the pressure vessel after burst failure.

**Figure 19 materials-18-02485-f019:**
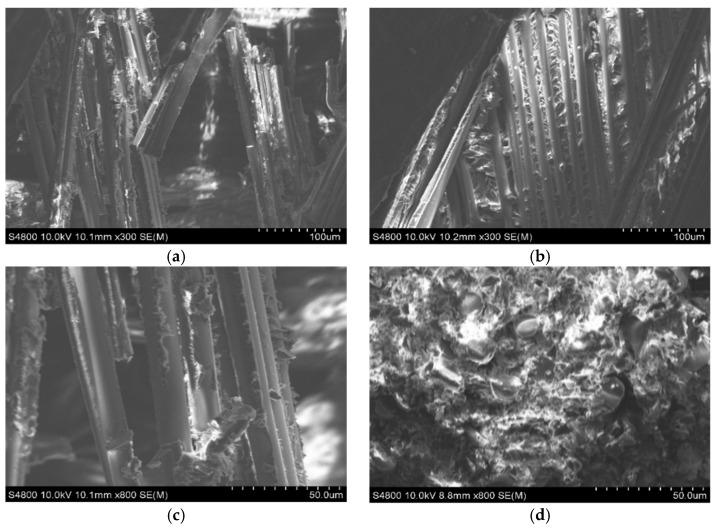
SEM of fracture surface of glass fiber epoxy resin composite pressure vessel: (**a**) fracture surface (0.1 mm); (**b**) after the fiber is soaked in glue (0.1 mm); (**c**) fracture surface (0.05 mm); (**d**) after the fiber is soaked in glue (0.05 mm).

**Table 1 materials-18-02485-t001:** Mandrel parameters.

Name	Barrel Diameter	Head Height	Diameter of the Front Pole Hole	Height of Rear Head	Diameter of Rear Pole Hole	Head Type
Unit (mm)	84.4	20	24	20	24	ellipsoid

**Table 2 materials-18-02485-t002:** Material parameter.

Bobbin Number	Single Bundle Yarn Width	Fiber Mass Content	Fiber Fineness	Fiber Density	Resin Density
1	5 mm	74%	1200 g/kg	1.8 g/cm^3^	1.2 g/cm^3^

**Table 3 materials-18-02485-t003:** Winding parameters.

Number	Number of Contact Points	Winding Number	Degree of Coverage %
1	13/5	67	107
2	−18/7	70	112
3	18/7	77	123
4	−23/9	64	102
5	23/9	74	118
6	21/8	76	122
7	−13/5	73	117
8	−28/11	79	126
9	28/11	89	142
10	−29/11	66	106
11	29/11	79	126
12	−21/8	71	114
13	−5/2	63	101
14	8/3	69	110
15	−8/3	69	107
16	5/2	70	107
17	27/10	89	142

**Table 4 materials-18-02485-t004:** Parameters of glass fiber composites.

Name	Unit/GPa
Young’s modulus E1	39
Young’s modulus E1	9
Poisson’s ratio	0.25
Shear modulus G12	4.35
Shear modulus G13	4.35
Shear modulus G23	2.2

**Table 5 materials-18-02485-t005:** Test scheme design.

Scheme Number	Specimen Number	Number of Spiral Winding Turns	Number of Toroidal Winding Layers	Whether the Head Is Coated with Glue
1	1	67	1	Yes
	2	67	1	No
	3	62	1	Yes
2	4	57	1	Yes
	5	52	1	Yes

## Data Availability

The original contributions presented in this study are included in the article. Further inquiries can be directed to the corresponding authors.

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
