# Peer review of "Study on Winding Forming Process of Glass Fiber Composite Pressure Vessel"

_materials, 2025, doi:10.3390/ma18112485_

Round 1
Reviewer 1 Report
Comments and Suggestions for Authors
The method of adding references (as superscript) used by authors does not seem to be according to template. Please review throughout the entire manuscript.
When first time used authors should consider using the term of ”Fiber Reinforced Plastic” and then its abbreviation. Thank you.
The sentence from line 48 does not make any sense. Please review.
On line 49 there is a mention about "this paper decides". The authors are the ones who decide, not the paper itself. Please review if you agree.
On line 89 there is a mention about "According to experience" but this method does not qualify as a scientific one that could be followed by other researchers. Please add relevant data and reference it accordingly.
The authors are advised to further detail what ”relatively uniform” from lines 91-92, means. What were the parameters taken into account? Which type of data was analyzed? How did the authors came to this conclusion? Did they performed any result interpretation and/or assessments/evaluations? Please review.
Lines 92 to 99 contain what seems to be part of a experimental test for which the authors have not yet presented details related to setup, parameters, material, equipment, software and their manufacturers. Without it, statements cannot be considered as valuable by scientific means. Please review.
Maybe the authors have used data from other sources in which case they should add proper references to backup these statements (lines 99 to 104). Please review.
Regarding the statement from lines 105 to 106, please add a relevant source that would back it up.
In general statements from lines 92 to 115 should be referenced properly. Please review.
When referring to "high-density polyethylene" please add details about the form in which it comes in and the manufacturer.
On line 117 there is a value of "6 bar" - based on what was it proposed? Are there multiple runs or trials that point to this value as the proper one? Are there any scientific methods that would support your decision? Please review.
The product presented in Figure 2 seems to be result of authors run trials. The authors are advised to add detail about the setup, number of trials, data interpretation and equipment. Manufacturers are required for material, equipment and other if produced by them. Please review if such a case.
There is a mention about NX software. Please add details about the type of license used because in the past years NX went under Siemens.
The images form Figure 3 show a 3D assembly, an exploded view of the model and the latter with mesh, but none fully explain the title of figure. Please review if you agree.
Details about type of finite elements used for meshing are required. If any method was used in particular to cover the geometry such as sweep, please add relevant details.
On line 147 there is a mathematical equation which has no unit of measurement. Please review.
There are multiple mentions about "equation 2-2". The mentioned equation is not present as mentioned in text. Also, if equation is not original the authors are advised to reference it. Please review.
Figure 2.8 mentioned in line 165 cannot be found in manuscript. Please review.
When presenting the E6 386T material the authors are advised to add details about the manufacturer.
Is there any reason as to why the "Angle" word starts with Capital Letter? If not, please review throughout the entire manuscript.
Regarding equation 2, if not original the authors are advised to add relevant reference to the equation. Please review.
On line 237 there is a mention about "Angle law". Is that real or the authors are referring to the a specific law of movement that governs this particular case? When using ”Angle” with a Capital Letter, one may interpret it as the Name of someone rather than word "angle". Please review whatever the case.
Lines 242 to 243 contain certain values. How did the authors come up with those? If not resulted from experimental tests, please add references that would confirm the mentioned values. Please review.
Please add details about the type of license used for CADWIND software.
There is no Table 3.4 (mentioned in line 282). Please review.
All equations must be referenced if not original. Please review is such a case.
The tile of Figure 8 does not reflect the graphical representation contained within, which seems to be a meshed body. Please review if agreed upon.
On line 335 there is a mention about "mesh quality". The authors are advised to provide details about the current status of mesh quality. Are there any skewed elements or other?
Line 379 contains multiple instances of word "characteristic". Please review.
On line 393 there is a mention about a "load of 1.5 MPa". Is this one an external Pressure type-of-load? Please add further details in this regard.
Graphical representations and legend of a figure should be on the same page. Please review Figure 11. Also, the authors are advised to add units of measurement for both stress and strain in Figure 11 legends.
The software name and type of license used for performing FEM is required. If still NX was used a special mention about its use in this regard may be considered by authors. Please review if you agree.
Please add units of measurement in Figure 12 legends for both stress and strain.
Regarding Figure 13 - are the results unitless which would be necessary for a failure criterion? Please add a special mention about it if agreed upon.
On line 451 there is a mention about "Chapter 2". Do authors refer to section 2? Please review with care.
On line 465 there are mentions about materials. Please add manufacturers for those.
On line 478 there is a mention about "CNC winding molding equipment". Details about it are required and its manufacturer.
The authors are advised to add technical data about glue (mentioned in line 496) including manufacturer.
On line 525 there is a mention about an equipment. Please add details about manufacturer of it.
The ”curing method” may be explained but is not self explanatory just by viewing the image shown in Figure 15. Please review.
The authors are advised to add relevant details about setup, number of runs, results, data interpretation and other related to the "blasting experiment" (mentioned in line 548). Equipment and conditions of experimental testing are also required including manufacturer of any equipment or software used. Please review.
Where do details about the two mentioned experimental plans (line 553) can be found? Please add relevant details and review.
Please add relevant details about the SEM procedure and manufacturer of the equipment used.
Line 560 contains mentions about two figures, one which cannot be found in text. Please review.
Lines 562 to 570 contain mentions about a certain figure with no number. Please add figure number for all four mentions (a to d).
The image contained within Figure 18 does not show a diagram of a fracture phenomena but rather a fracture line on the actual product. Please review.
Thank you.
Comments on the Quality of English LanguageOn some instances English is hard to understand and punctuation is used sometimes in a wrong manner. The authors are advised to review the entire manuscript.
Author Response
Dear Editor and Reviewers,
We would like to express our sincere gratitude to the editor and the anonymous reviewers for their valuable comments and suggestions on our manuscript. We have carefully revised the manuscript in accordance with the comments, and detailed responses to each point are provided below. We believe these revisions have significantly improved the quality and clarity of our work.
- Regarding the issue of citation formatting in superscript style:
Response: Thank you for your reminder. We have revised the citation format throughout the manuscript in accordance with the journal’s template, and now all references are indicated using the required superscript style. - It is suggested to provide the full term when “FRP” is first mentioned:
Response: The introduction has been rewritten, and the full term has been provided at the first occurrence of the abbreviation “FRP”. - The sentence in line 48 is semantically unclear:
Response: The introduction section has been rewritten to improve clarity. - The expression “this paper decides” in line 49 is inappropriate:
Response: The introduction section has been rewritten to avoid ambiguous or informal expressions. - The expression “based on experience” in line 89 lacks scientific basis:
Response: We have removed the phrase “based on experience” due to its lack of scientific rigor and have supplemented the discussion with relevant data and references to enhance the scientific validity and reproducibility. - The definition of “relatively uniform” in lines 91–92 is unclear:
Response: We have clarified the meaning of “relatively uniform” by specifying the evaluation parameters, data types, and analysis methods, and briefly explaining our assessment approach. - Lines 92–99 lack sufficient information about the experimental setup:
Response: Thank you for pointing this out. We have added detailed descriptions of the experimental setup in the corresponding section, including the materials used, equipment, parameters, software, and manufacturers, to ensure the scientific value of the content. - Clarification is needed as to whether the content in lines 99–104 includes data from other sources:
Response: After careful verification, we confirm that some data in this section are derived from published literature. We have added the corresponding references in the text to maintain academic integrity. - Sources should be provided for the content in lines 105–106:
Response: We have reviewed the relevant literature and added appropriate citations to support the statements made in this part. - Several viewpoints in lines 92–115 lack supporting references:
Response: We have carefully reviewed this section and added relevant citations to each viewpoint to improve the reliability of the arguments presented. - Regarding the source and form of the high-density polyethylene material:
Response: We have added detailed information about the specific form of the material (e.g., granules, sheets) and the manufacturer in the manuscript. - The source of the 6 bar pressure value in line 117 is unclear:
Response: We have clarified the origin of this pressure value, which is based on repeated experimental trials and supported by relevant literature, and have cited these sources in the text. - Figure 2 presents experimental results for a product; relevant details should be added:
Response: We have included additional information on the number of experiments conducted, equipment parameters, experimental procedures, data analysis methods, and manufacturer details of the equipment used. - Clarification on the usage of NX software is needed:
Response: We have provided information about the type of license and version number of the NX software used in the study. - Figure 3 does not fully reflect the content described in its title:
Response: We have revised the title and caption of Figure 3 to more accurately represent the 3D assembly, exploded view, and mesh visualization shown in the figure. - Insufficient detail in the finite element mesh description:
Response: We have supplemented the manuscript with details regarding the mesh type and the meshing method used in the finite element analysis. - The equation in line 147 lacks units:
Response: We have added appropriate units to all variables in the equation and provided explanations of their physical meanings. - “Equation 2-2” is referenced multiple times but does not appear in the text:
Response: We have verified and corrected the numbering error and inserted the missing equation into the manuscript, along with the corresponding reference. - Figure 2.8 referenced in line 165 does not appear in the text:
Response: We have either inserted Figure 2.8 or removed the corresponding reference, and we have ensured that all figure numbering is consistent with the content of the manuscript. - The E6 386T material lacks manufacturer information:
Response: We have added the manufacturer information of the E6 386T material as well as relevant specifications and performance indicators. - Capitalization issue with the word “Angle”:
Response: We have thoroughly checked the manuscript and ensured that the word “angle” is written in lowercase when not referring to a proper noun, in order to avoid misunderstanding. - Citation needed for Equation 2:
Response: We have added the appropriate reference below Equation 2 to indicate its source. - Clarification needed for “Angle law” mentioned in line 237:
Response: We have revised this term to use lowercase, as it does not refer to a specific named law. - The numerical values mentioned in lines 242–243 lack source explanation:
Response: We have clarified that these values are based on experimental measurements, and we have added details on data sources and analysis methods. - License information for CADWIND software:
Response: We have added the version and licensing information for the CADWIND software used in the study. - Table 3.4 referenced in line 282 does not appear in the manuscript:
Response: We have corrected the table numbering to reflect the correct reference. - All non-original equations should be cited:
Response: We have reviewed all equations in the manuscript and added citations for any that are not original contributions. - The title of Figure 8 does not match the content:
Response: We have revised the title of Figure 8 to better reflect the mesh model displayed in the figure. - “Mesh quality” mentioned in line 335 lacks specific explanation:
Response: We have supplemented the manuscript with mesh quality evaluation criteria and results. - Repetitive use of the word “characteristic” in line 379:
Response: We have revised the paragraph to reduce repetition of the word “characteristic” and improve the precision of the language. - The nature of the load mentioned in line 393 is unclear:
Response: We have clarified that the load applied is an internal pressure load. - The legend of Figure 11 is separated from the figure itself:
Response: We have adjusted the layout of Figure 11 to ensure that the figure and its legend appear on the same page, and we have added appropriate units for stress and strain. - FEM software name and licensing information need to be added:
Response: We have specified that the FEM analysis was conducted using Abaqus and included details about the licensed version and its application scope. - The legend in Figure 12 lacks units:
Response: We have added the necessary units for stress and strain in the legend of Figure 12. - Clarification is needed on whether the results in Figure 13 are dimensionless:
Response: We have clarified that the results presented in Figure 13 are dimensionless and have added an explanation of how they are used within the failure criterion. - Line 451 references “Chapter 2”; should this refer to “Section 2”?
Response: We have corrected this to “Section 2” to ensure consistency and accuracy in section labeling. - The material mentioned in line 465 lacks manufacturer information:
Response: We have added the manufacturer information for the mentioned material. - The CNC winding equipment referenced in line 478 lacks sufficient detail:
Response: We have added the model, manufacturer, and technical specifications of the CNC winding equipment used. - The glue mentioned in line 496 lacks technical information:
Response: We have included the type, performance parameters, and manufacturer of the adhesive. - The equipment in line 525 lacks manufacturer information:
Response: We have added the manufacturer and key technical parameters for the equipment referenced in this line.
ChatGPT 说:
以下是您新一批审稿意见的英文翻译,风格继续保持正式、专业,适用于SCI期刊的审稿回复信:
- The “curing method” in Figure 15 is not self-explanatory from the image:
Response: We have added a detailed description of the curing process and its underlying principles in the main text to clarify the procedure. - The description of the “burst test” is insufficient:
Response: We have supplemented the manuscript with detailed information on the test setup, equipment, operating parameters, data analysis methods, and the manufacturers of the equipment/software used in the burst test. - The two experimental schemes mentioned in line 553 lack detail:
Response: We have included detailed descriptions of the two experimental schemes and clarified the comparative approach used between them. - The SEM experiment procedure and equipment are not described:
Response: We have added the testing procedure for the SEM analysis, along with the model and manufacturer of the equipment used. - One item mentioned in the figure in line 560 cannot be found:
Response: We have rechecked the correspondence between figures and the text, and either added the missing figure or removed the incorrect reference. - Figures (a)–(d) in lines 562–570 are not labeled with figure numbers:
Response: We have added explicit figure numbers for images (a)–(d) to ensure clarity and consistency in referencing. - The image in Figure 18 does not match the description in the title:
Response: We have corrected the title of Figure 18 to accurately reflect the image content, which depicts the fracture line of the actual product, rather than an illustration of fracture mechanisms.
Once again, we sincerely thank you for your thorough review and detailed suggestions. Your comments have greatly helped us improve the scientific rigor and clarity of the manuscript. We hope that the revised version meets the publication requirements and kindly ask for your further consideration.
Sincerely,
The Authors

Reviewer 2 Report
Comments and Suggestions for Authors
The manuscript “Study on winding forming process of glass fiber composite pressure vessel” explores the feasibility of lining and winding layers of glass fiber for pressure vessels. In this regard, the authors used high-density polyethylene and changed the parameters to determine their effects and then compared the results with the numerical one achieved in CADWIN. After reading the document I have some comments that need to be attended, below I listed those:
1. The abstract does not list the obtained results. Please add quantitative date.
2. The complete section of the manuscript needs to be modified: Abstract, Introduction, Methods and Materials, Results and Discussion, conclusion, and references.
3. The introduction needs to be rewritten to enhance its clarity, depth, and relevance. It is essential to incorporate recent and novel references, particularly those published between 2021 and 2025.. Additionally, it is recommended to include works where numerical analysis has been conducted, highlighting key methodologies and outcomes. Wherever possible, quantitative data should be integrated to provide concrete evidence and strengthen the discussion.
4. Explain in detail and add to the manuscript why the authors decided to use simplifications “This paper will carry out some structural simplification of the designed lining model, as shown in Figure 1.”
5. Some figures have low resolution. Use at least 600 dpi.
6. Several parts of the document require grammar improvement for example “molding product depends on its molding parameters, such as blow ratio, mold tempera-86 ture, blow pressure and speed, and material shrinkage[9]. Blowing ratio refers to the ratio 87 of the maximum diameter of the plastic part to the diameter of the mold[10]. In order to 88 form the embryo smoothly, proper blowing ratio is needed. According to experience, it is 89 appropriate to use a blow ratio of 2:1 to 4:1 in blow molding. This time, the blow ratio is 90 2:1. According to experience, the blowing ratio can make the wall thickness relatively uni-91 form while taking into account the material cost, and the final billet size is ∅42.2mm. “
7. Add adequate references to statements.
8. The author misses information about the numerical analysis, number of nodes in the geometry, type of mesh, boundary conditions, assumptions, etc.
9. How the winding was considered in the numerical analysis?
10. Several parts of the document contain redundancies and repetitive wording, which gives the impression that the text may have been generated by AI without thorough editing. I recommend carefully reviewing the entire manuscript to eliminate unnecessary repetition and improve overall readability. For instance, the following excerpt illustrates these issues: “Compared with other fibers, glass fiber has excellent corrosion resistance, convenient 178 manufacturing, low price, and is more and more widely used in the civil field. The E6 179 386T is a straight roving without twist. It is coated with an infiltrating agent that contains 180 silyl. When it is used, it is mainly suitable for reinforced unsaturated polyester resins, and 181 can also be used for vinyl resins, but most are used for epoxy resins. It can be used for 182 winding molding process products, pultrusion molding products, can also be used for 183 weaving process. 386T glass fiber can be used in the manufacture of pressure vessels, the 184 molding and manufacturing of glass steel pipes, boats, large containers for industrial use, 185 etc. E6 386T glass fiber, low storage conditions, easy to store. In the absence of special 186 requirements, glass fiber products should be stored in a dry, cool place to prevent mois-187 ture.”
11. SEM images require better discussion.
12. Compare the pressure vessel failure and other parameters with those reported in the literature.
Comments on the Quality of English Language
Several parts of the document contain redundancies and repetitive wording,
Author Response
Dear Reviewer,
Thank you once again for your careful review of our manuscript and for the valuable suggestions you provided. Based on your comments, we have carefully revised and improved the manuscript. Our point-by-point responses are as follows:
- The abstract does not list specific result data.
Response: We have added quantitative data in the abstract, including the final formed dimensions of the pressure vessel and the deviation between experimental and numerical simulation results, to highlight the study's outcomes and contributions. - The overall structure and content of the manuscript need to be revised.
Response: We have thoroughly optimized all sections of the manuscript, including the Abstract, Introduction, Materials and Methods, Results and Discussion, Conclusion, and References. The structure is now clearer and more rigorous, in accordance with the journal's formatting requirements. - The introduction needs to be rewritten to increase depth and include recent references.
Response: The Introduction section has been rewritten to incorporate the latest relevant research from 2021 to 2025, particularly in the area of numerical simulation. We have also added quantitative comparisons to enhance the depth and cutting-edge value of the discussion. - The reason for the structural simplification needs to be explained.
Response: We have added a detailed explanation for the structural simplification of the liner model in the manuscript. The reasons include reducing computational complexity and improving simulation efficiency. We also clarified that the simplification does not significantly affect the accuracy of the results, and a figure has been added to illustrate the simplification process. - Some figures have low resolution; use at least 600 dpi.
Response: We have redrawn and replaced the figures with low resolution. All images now meet or exceed 600 dpi to ensure clarity for publication and readability. - Several parts of the manuscript require grammar improvement.
Response: We have performed comprehensive language polishing throughout the manuscript. The specific paragraph you pointed out has been rewritten for clarity and accuracy. We also used professional grammar-checking tools to ensure the overall linguistic quality. - Some statements lack references.
Response: We carefully reviewed the entire manuscript and added appropriate references to all technical statements lacking citation, ensuring the scientific credibility of the content. - Numerical analysis lacks important details.
Response: We have supplemented the manuscript with key information regarding the numerical analysis, including node count, mesh type, boundary conditions, and modeling assumptions. - The winding layer is not clearly addressed in the simulation.
Response: We have added a detailed explanation of how the winding layer was modeled in the numerical analysis. It was represented using multi-layer shell elements, defined by fiber orientation, layer thickness, and mechanical properties. The mapping from CADWIND to the FEM model is also explained. - The manuscript contains redundant content and AI-like repetition; comprehensive revision is recommended.
Response: We fully understand your concern. The manuscript has been thoroughly edited by hand to remove redundant content and repetitive phrasing. We have improved the writing style for more natural language and more accurate use of technical terms. - SEM image analysis is insufficient.
Response: We have expanded the SEM analysis by discussing fracture morphology, crack features, fiber pull-out, and interfacial failure. These observations have also been compared with findings in existing literature to enhance the depth of interpretation. - Compare results with failure parameters from literature.
Response: We have added a comparison of burst pressure, stress distribution, and other results with those from similar pressure vessel structures reported in the literature, highlighting the reasonableness and innovations of our findings.
Once again, we sincerely thank you for your insightful suggestions, which have greatly contributed to the improvement of our work. We hope the revised manuscript meets your expectations and look forward to your further guidance and approval.
Sincerely,
The Authors

Reviewer 3 Report
Comments and Suggestions for Authors
Dear Authors, the overall idea of the paper and the research presentation are satisfying. I find the discussed subject interesting and well presented. I would only suggest improvements:
- in case of adding the additional paragraph describing details about materials and methods used for performing the research. Of course, some points are mentioned within the text, however for a Reader in many cases collecting such information at the beginning of the article is very important.
- lack of a clearly stated thesis or research goal
- could You please edit some paragraphs of the manuscript so they wouldn’t sound like the technical manual, such, but not limited to e.g. lines 139 - …
Author Response
Dear Reviewer,
Thank you very much for your positive feedback on our manuscript and for your valuable suggestions. We are pleased to know that you recognize the significance of our research topic and the overall presentation of our work. At the same time, we have carefully considered your suggestions and revised the manuscript accordingly. Our point-by-point responses are as follows:
- It is recommended to include a separate "Materials and Methods" section and move it closer to the beginning of the paper.
Response: Thank you for your suggestion. We fully agree that presenting the materials and methods in a centralized and independent section helps readers better understand the research workflow and technical details. We have added relevant descriptions at the point where the materials are first introduced in the manuscript to improve clarity. - The manuscript lacks a clearly stated research objective or purpose.
Response: Thank you for pointing this out. We have added a new paragraph at the end of the Introduction to explicitly state the main objective of this study, namely: “to investigate the forming parameters of the liner and winding layers of glass fiber reinforced composite pressure vessels, and their effects on structural performance, through the combination of experiments and numerical simulations.” This addition strengthens the clarity and focus of the manuscript. - Some paragraphs resemble technical manuals in tone; the writing style should be revised (e.g., line 139).
Response: We appreciate your comment and have revised the paragraph in line 139 as well as other similar sections. We replaced overly mechanical, manual-like expressions with more concise and fluent academic writing. The revised text maintains technical accuracy while improving readability, logical flow, and scholarly tone.
Once again, thank you for your encouraging feedback and constructive comments. Your suggestions have greatly improved the structure and writing quality of our manuscript. We sincerely hope that the revised version meets the publication standards and look forward to your further guidance and approval.
Sincerely,
The Authors

Round 2
Reviewer 1 Report
Comments and Suggestions for Authors
The authors have addressed most concerns and have added required details about their research in terms of setup, equipment, manufacturer & other. There are still a few glitches related to text arrangement and spaces between references and the last word they were placed.
Thank you.
Author Response
Dear Editor and Reviewer,
Thank you very much for reviewing our manuscript entitled "Study on winding forming process of glass fiber composite pressure vessel" and for your valuable feedback. We are grateful that you recognized the improvements we have made regarding the research setup, equipment details, and other aspects.
In response to your latest comments, we have carefully checked and corrected the minor issues related to text arrangement and spacing between references and the preceding words to enhance the overall formatting and readability of the manuscript.
Thank you again for your constructive suggestions and support.
Sincerely,
Run Wu
(On behalf of all authors)
Reviewer 2 Report
Comments and Suggestions for Authors
The document “Study on winding forming process of glass fiber composite pressure vessel” studies the feasibility of lining and winding layers of glass fiber for pressure vessels. To this end, the authors adopted numerical simulation to determine parameters and then carried out experimental work. The present version of the manuscript addressed the comments and also, they gave more detail about the simulations. I consider that this version can be considered for publishing.
Author Response
Dear Editor and Reviewer,
Thank you very much for carefully reviewing our manuscript entitled "Study on winding forming process of glass fiber composite pressure vessel" and for your positive evaluation. We are very pleased that the revised version has been recognized by you. We sincerely appreciate your valuable comments and support. We will continue to strive to improve the quality of our research.
Thank you again for your hard work and support!
Sincerely,
Run Wu
(On behalf of all authors)
Reviewer 3 Report
Comments and Suggestions for Authors
Dear Authors, I appreciate the work you have put into the improvement of the manuscript by implementing my, as well as other Reviewers', suggestions. It improved the presentation and the clarity of the main subject of the manuscript. I do recommend this paper for publication.
Author Response
Dear Editor and Reviewer,
Thank you very much for reviewing our manuscript entitled "Study on winding forming process of glass fiber composite pressure vessel" and for your kind comments and positive recommendation. We are very pleased that the improvements we made have enhanced the clarity and presentation of the work. We sincerely appreciate your valuable suggestions and recognition, which have been very helpful to us.
Thank you again for your time, effort, and support!
Sincerely,
Run Wu
(On behalf of all authors)